# Effects of Gestational Age and Early Parenting on Children’s Social Inhibition at 6 Years

**DOI:** 10.3390/children6070081

**Published:** 2019-06-28

**Authors:** Lucia M. Reyes, Julia Jaekel, Dieter Wolke

**Affiliations:** 1Department of Child and Family Studies, The University of Tennessee, Knoxville, TN 37996, USA; 2Department of Psychology, University of Warwick, Coventry CV47AL, UK

**Keywords:** temperament, shyness, disinhibition, preterm birth

## Abstract

Preterm birth (<37 weeks’ gestation) has been associated with problems in social functioning. Whether social inhibition is specifically related to preterm birth and whether early parenting may protect against social inhibition difficulties is unknown. To explore effects of gestational age and early parent–infant relationships on social inhibition, 1314 children born at 26–41 weeks gestational age were studied as part of the prospective Bavarian Longitudinal Study. Early parent–infant relationship quality was assessed postnatally with the parent–infant relationship index. Social inhibition was assessed at age 6 years using an experimental procedure, in which nonverbal and verbal responses were coded into social inhibition categories (disinhibited, normally responsive, inhibited). Multinomial logistic regressions indicated that children with lower gestational age showed more socially disinhibited (nonverbal: OR = 1.27 [95% CI = 1.17–1.40], verbal: OR = 1.23 [95% CI 1.13–1.35]) and inhibited (nonverbal: OR = 1.21 [95% CI = 1.11–1.32], verbal: OR = 1.11 [95% CI = 1.01–1.21]) responses. Good early parent–infant relationships were associated with less verbal disinhibition (OR = 0.70 [95% CI = 0.52–0.93]). Findings suggest that children with lower gestational age are at greater risk to be both socially inhibited and disinhibited. Early parenting affected risk of abnormal social responses. Supporting early parent–infant relationships may reduce preterm children’s risk for social difficulties.

## 1. Introduction

Temperamental characteristics predict later socioemotional and behavioral problems [1]. Infants’ and toddlers’ behavioral inhibition—the temperamental tendency to react to novel stimuli with wariness and avoidance [2]—for example, is a precursor to social reticence [3] and predictive of anxiety in middle childhood and adolescence [4,5,6]. Social inhibition, which specifically refers to responses to unfamiliar social stimuli, is more closely associated with later socially anxious behaviors than inhibition towards nonsocial stimuli [7,8]. In contrast to social disinterest, social inhibition reflects emotional dysregulation characterized by high fear in low-threat situations [9]. In young children (2–6 years), social inhibition is assessed via the quality and timing of responses to an adult stranger [7,10,11]. Efforts to identify an early-life at-risk phenotype for social anxiety have focused on neurobiological correlates of inhibited temperament and suggest that specific brain circuits underlie its behavioral manifestation [12]. However, the role of perinatal risk on the development of social inhibition is not well understood [13]. 

Preterm birth (<37 weeks gestation) is known to cause brain injury and alterations [14] that may be related to the development of inhibited temperament [15], thus putting preterm children at risk for social difficulties [16]. Compared to their full-term (39–41 weeks) peers, very preterm (<32 weeks) children show more peer problems, greater social withdrawal, and poorer social skills [16,17,18], which may increase their risk of social exclusion and being bullied by peers [19]. These difficulties appear to be long-lasting, as extremely preterm (<28 weeks gestation) individuals have more peer problems and lower confidence in adolescence [20], and more shyness and lower sociability in early adulthood [21,22]. Very preterm adults are also at greater risk for low social support [23,24] and a withdrawn personality type [25,26]. Although early temperamental traits may underlie these difficulties, previous studies have not used experimental paradigms to assess preterm children’s social inhibition.

Further, whereas very preterm children have a known greater risk of social problems than term-born peers, less is known about children born moderately (32–33 weeks gestation) and late preterm (34–36 weeks gestation; [27]). The scarce evidence suggests that they might also be at risk for socioemotional problems compared to full-term counterparts [28]. For instance, mothers of late preterm infants rate their children more highly on temperamental negativity than do mothers of full-term infants [27]. By school age, late preterm children have more internalizing and externalizing problems, lower IQ, more school problems [29,30], and fewer friends [31] than full-term peers. Thus, temperamental characteristics (i.e., social inhibition) might offer a potential pathway to the subsequent social difficulties observed in individuals born very preterm, but whether these effects gradually increase with neonatal risk across the full range of gestation remains unknown.

Although positive early parent–infant relationships are important for all children, their contribution may be especially protective for preterm children’s social development. High-quality parent–child interactions have been positively associated with preterm children’s early social–emotional competence [32,33,34], particularly for those with higher medical risk [35,36]. Children born with very low birth weight (<1500 g) are more vulnerable than normal birth weight children to the effects of parental depression [37] and insensitive parenting [38,39]. Less sensitive parenting predicts impairments in self-regulation in very preterm toddlers [40], while maternal anxiety and negative or intrusive parenting predict poorer social competence at age 4 years of age [17].

Therefore, using data from a larger longitudinal study, the current study aimed to investigate the impact of gestational age and parent–infant relationships on social inhibition across the full gestation range. The current research questions were developed after the data collection had been finalized in the larger study. It was hypothesized that lower gestational age would predict higher social inhibition, and that good early parent–infant relationships would predict lower social inhibition. 

## 2. Materials and method

### 2.1. Participants

Participants were recruited as part of the larger prospective, geographically defined, whole population Bavarian Longitudinal Study of neonatal at-risk children in Germany. All infants born between January 1985 and March 1986 admitted to a children’s hospital within the first 10 days of life (*n* = 7505; 10.6% of all live births), and 916 healthy term control infants born in the same hospitals during the same period, who received normal postnatal care, were recruited into the study and assessed at birth and 5 months of age. At age 6 years, *n* = 1314 of the initial sample, stratified by sex, socioeconomic status (SES), and degree of neonatal risk, were selected for follow up [38,41]. Ethical approval was granted by the Ethics Committee of the University of Munich Children’s Hospital and the Bavarian Health Council (Landesärztekammer). Parents provided written informed consent to participate in the study within 48 h of their child’s birth. Further details of the study design can be found elsewhere [38,41].

### 2.2. Measures

**Biological and sociodemographic variables at birth**. Gestational age, birth weight, and sex were obtained from obstetric records. Family SES at birth was based on maternal and paternal education and occupational status and coded from 1 (lowest) to 6 (highest social class) [42], then reverse coded for analyses.

**Parent–infant relationship**. Early parent–infant relationship quality was assessed with the parent–child relationship index (PIRI) from birth to five months [43,44]. The instrument consists of eight yes/no items obtained from trained nurses’ observations and a standardized interview with the infants’ parents, assessing attachment-related parental concerns, feelings, and behaviors, with items such as: *mother shows little pleasure when interacting with the child* (nurse’s observation, neonatal), and *mother has difficulties in establishing a relationship to the infant* (mother interview, at 5 months of age). After all items were obtained, the sum of responses was calculated and recoded into one binary variable (0 = no concerns, 1 = some degree of concern). Further details of this assessment are described in Appendix A (see also [43]). 

**Social inhibition assessment.** At age 6 years, children’s social inhibition in an unfamiliar setting was assessed with a standardized experimental procedure of the child’s interaction with an adult stranger [10]. The task was designed to specifically test children’s temperamental social inhibition and had high inter-rater (93–94) and retest (0.75 between ages 4 and 6 years) reliability in the original study [10]. Children and their mothers were placed in a specially equipped soundproof room without toys. The mother was seated 1.5 to 2 m away from the child and instructed to answer a written questionnaire and not actively engage with the child. When the child started to show signs of being bored (between 2 to 5 min after entering the room), an adult stranger entered with a transparent bag filled with toys, greeted child and mother, and sat down opposite of the mother about 1 m from the child. The stranger then started unpacking the bag of toys and playing with them while looking at the child every ten seconds but not actively approaching the child. If the child had not initiated nonverbal (e.g., pointing to toy and looking at stranger) or verbal (e.g., asking for toy; saying hello) contact after 180 s had passed, the stranger asked if (s)he wanted to play with the toys. The number of seconds that passed before the child initiated verbal or nonverbal contact (either before or after the stranger’s cue) was recorded with a stopwatch. Based on the distribution of responses for the healthy (i.e., not neonatally hospitalized) full-term children (Figure 1 and Figure 2), latencies of all children’s nonverbal and verbal responses to the stranger were coded into 3 social approach categories: 1 = disinhibited (response nonverbal: <180 s; verbal: <180 s), 2 = normally responsive (response nonverbal: 180–188 s; verbal: 180–227 s) and 3 = inhibited (response nonverbal: >188 s, verbal >227). This response pattern corresponded with the experimental stimulus (i.e., the stranger inviting the child to play after 180 s); thus, children were coded as not socially inhibited or disinhibited if they showed the expected developmentally-appropriate response to the social cue. Further details for the coding of this assessment are provided in Appendix A. 

### 2.3. Statistical Analysis

All analyses were performed using SPSS v. 24 (Chicago, IL, USA). Mean values and frequencies for descriptive characteristics were calculated and are reported by gestational age group in Table 1. Multinomial logistic regressions were performed to determine the impact of higher gestational age and PIRI scores on social inhibition behavior compared to the ‘normal response’ (nonverbal: 180–188 s; verbal: 180–227 s), controlling for sex and SES. Regressions were performed for both verbal and nonverbal responses with gestational age group (i.e., 1 = full term, 2 = early term, 3 = late preterm, 4 = moderately preterm, 5 = very preterm) and PIRI (i.e., 0 = no concern, 1 = some concern) as predictors of interest. In addition to main effects, the interaction of gestational age group with PIRI scores was tested. Finally, multinomial regressions were used to calculate the relative odds of verbal inhibition and disinhibition for each gestational age group compared to the full-term group.

## 3. Results

Table 1 outlines descriptive characteristics of the final sample according to gestational age group status. Preliminary analyses showed no statistically significant differences in sex across groups, but lower gestational age was associated with lower SES and poorer PIRI scores. Frequencies of the social inhibition assessment results by gestational age groups are presented in Table 2. Children of lower gestation were not only more likely to show socially inhibited but also disinhibited verbal and nonverbal responses than those of higher gestational age groups.

Results for multinomial logistic regressions are displayed in Table 3 and Table 4. Children born with lower gestational age had higher odds of exhibiting socially inhibited (nonverbal: OR = 1.21 [95% CI = 1.11–1.32], verbal: 1.11 [95% CI = 1.01–1.21]) response behavior. Thus, *on average*, each lower gestational age group had 21% increased odds of nonverbal inhibition and 11% increased odds of verbal inhibition than the next higher gestation group, controlling for sex, SES, and parent–infant relationship quality. Additionally, children of lower gestational age groups were more likely to show socially disinhibited responses (nonverbal: OR = 1.27 [95% CI = 1.17–1.40], verbal: OR = 1.23 [95% CI = 1.13–1.35]), meaning that on average, each lower gestational age group had 27% increased odds of nonverbal disinhibition and 23% increased odds of verbal disinhibition compared to the next gestational age group.

Good early parent–infant relationship quality was not associated with differences in verbal nor nonverbal inhibition, but with lower verbal disinhibition (OR = 0.70 [95% CI = 0.52–0.93]), after controlling for child sex, family SES, and gestational age group. In other words, children with a good PIRI score had 30% decreased odds of having verbally disinhibited responses. Because there was not a significant interaction effect of parent–infant relationship with gestational age on social inhibition, results reported above are only for the more parsimonious models without the nonsignificant interaction term. Results for the models with the interaction term can be found in Appendix A. Frequencies of the social inhibition assessment verbal results by PIRI score are presented in Table 5. 

Results of multinomial regressions comparing verbal responses for each gestational age group to the full-term group are shown in Table 6, and odds ratios are presented in Figure 3 for social inhibition and Figure 4 for disinhibition.

## 4. Discussion

This study investigated effects of early parent–infant relationship quality and preterm birth across the whole gestational age range on children’s social inhibition at 6 years. It was found that children with lower gestational age had higher odds of being socially inhibited and disinhibited than children with higher gestational age. In addition, good parent–infant relationship quality was associated with less verbal disinhibition.

These findings add to the existing literature suggesting that both gestational age [15] and early parenting [45] impact children’s approach in unfamiliar social situations. New is that this study identified that both ends of temperamental approach tendencies may underlie social difficulties in preterm children; being either inhibited (withdrawn) or disinhibited may put preterm children at higher risk of social rejection in relationships. Further, this study adds to existing literature on the importance of early parenting in protecting against difficulties for children across the whole gestational age range [33]. Studies have documented that social problems in very preterm individuals prevail into adolescence and adulthood [20,26,46], and results of the current study illuminate a potential underlying temperamental pathway, evident in early childhood. Previous work has also suggested that prematurity might impact the development of temperament [47], and the current study identifies social inhibition as a more precise temperamental trait with extremes of inhibition and disinhibition that are more prevalent with lower gestation at birth and might help to explain the persistence and trajectory of social problems. Temperament-based inhibition towards unfamiliar social stimuli can be distinguished from inhibition due to social–evaluative concerns (see [10]). Our study explored the former and suggests that prematurity is related to inhibition towards unfamiliarity. Interestingly, these findings do not only point to increased social inhibition but also disinhibition as a potential precursor of later social behavior problems following preterm birth. Both deviations from normal approach behavior may put preterm children at risk for adverse social interactions such as bullying [19,48], adding to social difficulties preterm children may have. Additionally, the current findings extend the emerging literature on social problems among moderate and late preterm individuals [27,29,49], underscoring the importance of considering degree of prematurity [50].

To our knowledge, this is the first investigation of the potential neurodevelopmental mechanisms underlying children’s social difficulties across the full range of gestation. The findings are suggestive of a dose–response effect of low gestational age on the ability to inhibit and disinhibit behavioral responses to new social situations. One potential mechanism for this social dysregulation among preterm children may involve structural and functional brain alterations [51,52]. A strong body of literature has demonstrated the role of brain connectivity in supporting emotional signaling and affect sharing [53,54,55,56], which are foundational to social engagement [57] and, more generally, to what is known as the “social brain”—the network of brain areas involved in recognition and evaluation of others and their mental states [58]. Preterm children are at risk for brain injury and neuronal/axonal disease [14], which may alter the structure and subsequent typical maturation of certain brain regions [59], including those involved in processing faces, detecting and responding to social cues, evaluating emotion, and processing others’ actions (for a review, see [51]). 

During the first months of life, preterm infants have greater difficulty orienting to social stimuli and maintaining social interactions [60,61] and positive affect when attending to faces [62,63,64]. A recent study showed that infants born preterm ( >35 weeks) with compromised brainstem functions during the perinatal period were at greater risk to display difficulties in regulating gaze during face-to-face interactions at 4 months than healthy preterm infants [13], suggesting that humans may be programmed for social interactions before experiencing social encounters. Further supporting this hypothesis, a subsequent study of this sample found that the risk of compromised brainstem functions involved in behavioral inhibition at 12 months was moderated by gaze engagement at 4 months [15]. While only a few studies have investigated associations of structural brain alterations with socioemotional outcomes in preterm individuals [52,65,66], the current evidence in this area also supports a neurodevelopmental pathway to social difficulties [51]. 

On the other hand, the etiology of disinhibited social behavior remains less clear. In the temperament literature, an abnormally high approach to novel social stimuli, including strangers, has been referred to as exuberance [67,68], or behavioral disinhibition [69], although it is acknowledged that the construct involves aspects previously regarded as high approach [70], high novelty seeking, and low harm avoidance [71]. However, preterm children and adults have been consistently found to show low novelty seeking [52,72,73], suggesting that the avenue to social disinhibition may be either related to altered brain development or early adverse experiences [74] and hospitalization or both. Most of these perspectives, nonetheless, recognize the role of the behavioral activation system (BAS), or motivation in disinhibition, and it is suggested that problems in inhibitory control might underlie social disinhibition [69,75]. Preterm children have been consistently shown to have problems in self-control abilities and inhibiting unwanted responses [40,76], including poorer control of regulation of behavioral states [77]. Poor self-control may manifest as difficulty delaying approach to a stranger when motivated by a stimulus (i.e., toys in the current study), despite social appropriateness or safety. Future studies should include BAS and inhibitory control measures to explore potential moderators of disinhibited social behavior in preterm children, given that highly disinhibited behavior is also linked with later emotional and behavioral problems [78]. 

Another possibility is that the social disinhibition observed here may share etiological pathways with attachment formation, as is conceptualized with indiscriminate approach behavior in disinhibited social engagement disorder (DSED). While preterm children, in general, are not more often insecurely attached than their term-born peers, they do have higher rates of disorganized attachment [79]. However, preterm children’s mothers are as sensitive as full-term children’s mothers [80]. Differences in preterm children’s attachment patterns could be partly attributable to the NICU experience, in which long hospitalization and incubator care limit contact with parents (e.g., touch, eye contact) and may hinder parent–child relationships [81,82,83]. Moreover, multiple painful procedures administered in the NICU may program the hypothalamic–pituitary–adrenal (HPA) axis for heightened reactivity [84]. Studies have not only shown that preterm infants are more likely to show dysregulated early behavior, such as crying, sleeping, and feeding problems [77] despite high-quality parenting [80], but that these early regulatory problems in preterm infants are more predictive of disorganized attachment than sensitive parenting [79]. Thus, early environmental factors beyond parenting (e.g., NICU, pain) as well as within child factors (e.g., brain injury, impaired neurodevelopment) and early regulatory problems should be considered as potential contributions to disinhibition. 

Importantly, results of the current study indicate a protective effect of good parent–infant relationships against social dysregulation. In contrast to our expectations, there was not a significant interaction of gestational age and parent–infant relationship quality on social inhibition. It should be noted that interactions are often difficult to detect due to lack of statistical power; in this study, a nonsignificant trend showed greater importance of parent–infant relationship quality for very preterm and moderately preterm children’s outcomes (Table 5). Overall, a good parent–infant relationship is protective for all infants, including those with high neonatal risk, from showing disinhibited verbal social behavior. There is a good case to assess the effectiveness of parent–infant interventions that have shown promise for preterm infant self-regulation [85]. 

Strengths of this study include its large, prospective, whole-population design, followed longitudinally until age 6 years, and its inclusion of control variables. In contrast to studies that rely on parent or teacher reports to assess children’s temperamental traits [47,86,87,88], a standardized experimental observation measure of social inhibition was used [10]. To assess the quality of parent–child relationships, trained nurses’ observations and parents’ reports were combined into one score (i.e., the PIRI). Despite its ecological validity, limitations of the PIRI include its lack of inter-rater reliability information and the potential subjectivity of parent self-reports. Additional limitations of the study include sample recruitment between 1985 and 1986, which warrants replication with more recent samples. Nevertheless, comparisons of older with more recently born cohorts’ long-term developmental outcomes suggest that, despite improved survival, neurodevelopmental outcomes have shown no change for the better over the last decades [41,89,90,91,92]. Since the first assessment of social inhibition in this study was conducted at six years, it was not possible to identify continuity in temperament from birth to age six, although temperamental traits are regarded as fairly stable across development [93]. Future studies could consider coding additional indicators of temperamental approach tendencies (e.g., emotional expressions), in addition to social response time during experimental tasks of children’s interaction with adult strangers. 

## 5. Conclusions

Results of this study add to emerging evidence of a dose–response effect of low gestation on children’s social inhibition. Health care professionals can provide regular follow-up assessments of children born preterm to support screening and early identification of potential social competence problems [94]. Future interventions to improve social interaction skills should start before school entry to prevent long-term problems [95] and should emphasize the role of parents in promoting the social development of their children, especially for those born very and moderately preterm. Additionally, future studies that explore social inhibition as a developmental pathway to preterm individuals’ later psychopathology [96] and social problems [97] constitute an important direction to shed light on the mechanisms explaining preterm individuals’ long-term outcomes.

## Figures and Tables

**Figure 1 children-06-00081-f001:**
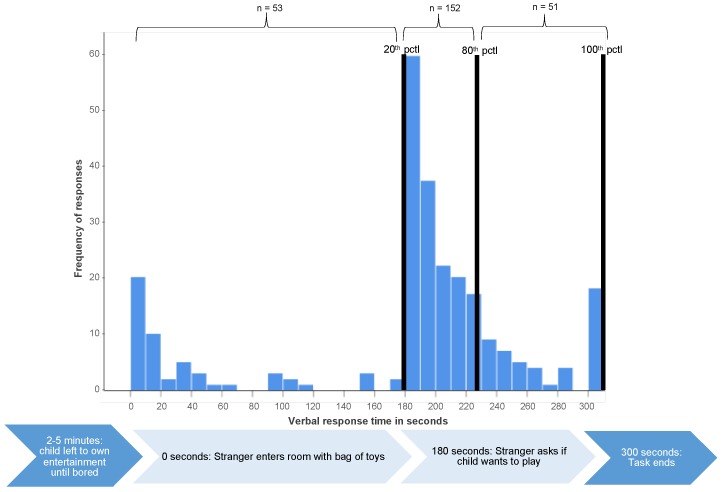
Distribution of healthy (i.e., non-neonatally hospitalized) full-term control children’s (*n* = 256) latencies of verbal response on social inhibition assessment. Note: Response time for children that showed no social reaction by 300 s (end of the task) was coded as 301 s. Pctl = percentile.

**Figure 2 children-06-00081-f002:**
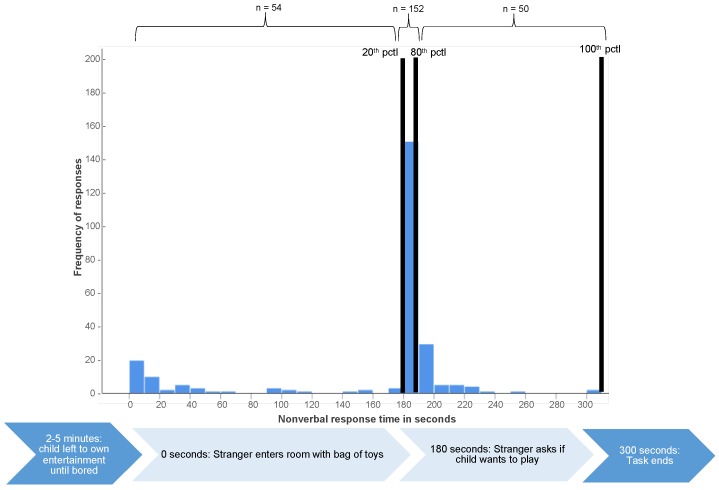
Distribution of healthy (i.e., non-neonatally hospitalized) full-term control children’s (*n* = 256) latencies of nonverbal response on social inhibition assessment. Note: Response time for children that showed no social reaction by 300 s (end of the task) was coded as 301 s. Pctl = percentile.

**Figure 3 children-06-00081-f003:**
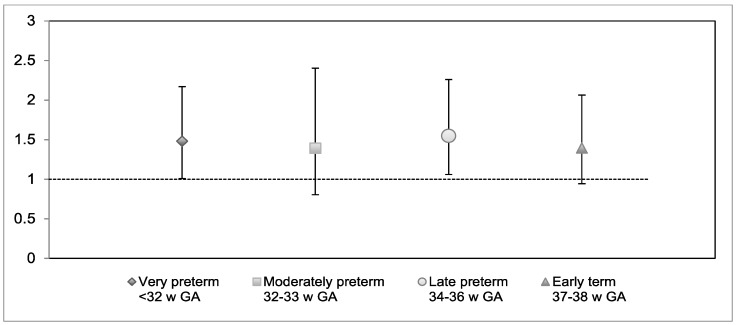
Odds ratios (OR) for verbal inhibition by gestational age group compared to the full-term group, controlled for sex and socioeconomic status. *Note*. *w GA* = weeks of gestation. Error bars denote 95% confidence intervals. There was a significant effect of lower socioeconomic status (OR: 1.18 [95% CI: 1.08–1.29]) on increased verbal inhibition.

**Figure 4 children-06-00081-f004:**
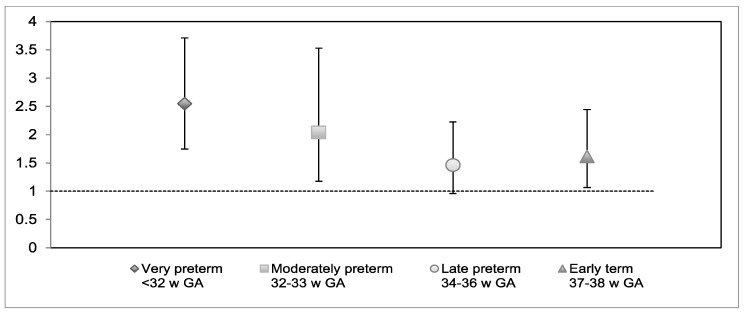
Odds ratios (OR) for verbal disinhibition by gestational age group compared to the full-term group, controlled for sex and socioeconomic status. *Note*. *w GA* = weeks of gestation. Error bars denote 95% confidence intervals. There was a positive effect of male sex (OR: 1.49 [95% CI = 1.13–1.98]) and a negative effect of lower socioeconomic status (OR: 0.89 [95% CI = 0.81–0.97]) on increased disinhibition.

**Table 1 children-06-00081-t001:** Descriptive background characteristics by gestational age groups (*n* = 1314).

Background Characteristics	Very Preterm 32 w GA *n* = 229	Moderately Preterm 32–33 w GA *n* = 90	Late Preterm 34–36 w GA *n* = 205	Early Term 37–38 w GA *n* = 199	Full Term 39–41 w GA *n* = 591	*F/X^2^* (df), *p* Value
Gestational age in weeks M(SD)	29.5 (1.5)	32.5 (0.5)	35.1 (0.8)	37.6 (0.5)	40.0 (0.7)	*F* (4,1309) = 6687.2 ***
Birth weight in grams M(SD)	1,287(323)	1,640 (375)	2,219 (547)	2,827 (559)	3,407 (504)	*F* (4,1309) = 951.3 ***
Child sex (% male)	57.2%	47.8 %	50.2%	48.2%	50.1%	*X^2^* (4,1314) = 4.7
SES (1 = high, 6 = low) M(SD)	3.5 (1.5)	3.5 (1.6)	3.4 (1.6)	3.3 (1.6)	3.4 (1.5)	*F* (4,1309) = 0.7
Good parent–infant relationship	50.7%	58.9%	62.9%	66.3%	70.1%	*X^2^* (4,1314) = 28.7 ***

*Note*. *w GA* = weeks of gestation. *SES* = socioeconomic status. Data are presented as *Mean (Standard Deviation)* for interval scaled and *percentages* for categorical variables. *** *p* > 0.001.

**Table 2 children-06-00081-t002:** Raw frequencies of nonverbal and verbal inhibition by gestational age group (*n* = 1314).

	Very Preterm 32 w GA *n* = 229	Moderately Preterm 32–33 w GA *n* = 90	Late Preterm 34–36 w GA *n* = 205	Early Term 37–38 w GA *n* = 199	Full Term 39–41 w GA *n* = 591	*X^2^*(df), *p* Value
**Nonverbal Inhibition** ^a^						41.26(8) ***
Disinhibited (%)	32.30%	27.80%	21.50%	24.60%	17.40%	
Normal response (%)	41.00%	42.20%	53.70%	51.30%	62.40%	
Inhibited (%)	26.60%	30.00%	24.90%	24.10%	20.10%	
**Verbal Inhibition** ^b^						30.37(8) ***
Disinhibited (%)	31.40%	26.70%	21.00%	23.10%	17.10%	
Normal response (%)	43.70%	47.80%	50.20%	50.80%	60.40%	
Inhibited (%)	24.90%	25.60%	28.80%	26.10%	22.50%	

Note. *w GA* = weeks of gestation. ^a^ The following categorization was used for nonverbal responses: <180 s = disinhibited; 180–188 s = normal; >188 s = inhibited. ^b^ The following categorization was used for verbal responses: <180 s = disinhibited; 180–227 s = normal; >227 s = inhibited. *** *p* > 0.001.

**Table 3 children-06-00081-t003:** Multinomial logistic regression results for variables predicting nonverbal response (*n* = 1314).

	Variable	*B*	SE *B*	Wald	df	OR	95% CI
Disinhibited 180 s	Intercept	−0.86 ***	0.24	13.03	1		
	Sex (male)	0.27	0.14	3.50	1	1.31	0.99–1.73
	Lower SES	−0.18 ***	0.05	15.19	1	0.84	0.77–0.92
	Lower gestational age	0.24 ***	0.05	27.77	1	1.27	1.17–1.40
	Good parenting	−0.27	0.15	3.33	1	0.77	0.57–1.02
Inhibited >188 s	Intercept	−1.04 ***	0.24	18.55	1		
	Sex (male)	−0.36 **	0.14	6.79	1	0.70	0.53–0.91
	Lower SES	−0.02	0.05	0.27	1	0.98	0.90–1.07
	Lower gestational age	0.19 ***	0.05	17.09	1	1.21	1.11–1.32
	Good parenting	0.01	0.15	0.00	1	1.01	0.76–1.35
	*LR χ^2^*	70.09 ***					

*Note:* ** *p* < 0.01, *** *p* < 0.001. The reference category is “normal” (180–188 s) nonverbal response.

**Table 4 children-06-00081-t004:** Multinomial logistic regression results for variables predicting verbal response (*n* = 1314).

	Variable	*B*	SE *B*	Wald	df	OR	95% CI
Disinhibited 180 s	Intercept	−0.99 ***	0.24	16.91	1		
	Sex (male)	0.40 **	0.05	7.80	1	1.50	1.13–1.98
	Lower SES	−0.13 **	0.05	7.80	1	0.88	0.81–0.96
	Lower gestational age	0.21 ***	0.05	20.75	1	1.23	1.13–1.35
	Good parenting	−0.36 *	0.15	6.02	1	0.70	0.52–0.93
Inhibited >227 s	Intercept	−1.45 ***	0.25	34.76	1		
	Sex (male)	0.01	0.14	0.00	1	1.01	0.77–1.31
	Lower SES	0.16 **	0.05	11.68	1	1.17	1.07–1.28
	Lower gestational age	0.10 *	0.05	4.99	1	1.11	1.01–1.21
	Good parenting	−0.19	0.14	1.73	1	0.83	0.63–1.10
	*LR χ^2^*	68.87 ***					

*Note:* * *p* < 0.05, ** *p* < 0.01, *** *p* < 0.001. The reference category is “normal” (180–227 s) verbal response.

**Table 5 children-06-00081-t005:** Raw frequencies of verbal inhibition by gestational age group and parent–infant relationship index (PIRI).

	Very Preterm 32 Weeks GA *n* = 229		Moderately Preterm 32–33 Weeks GA *n* = 90		Late Preterm 34–36 Weeks GA *n* = 205		Early Term 37–38 Weeks GA *n* = 199		Full Term 39–41 Weeks GA *n* = 591	
*PIRI*	*Poor*	*Good*	*Poor*	*Good*	*Poor*	*Good*	*Poor*	*Good*	*Poor*	*Good*
Disinhibited (%)	36.3%	26.7%	35.1%	20.8%	19.7%	21.7%	28.4%	20.5%	18.1%	16.7%
Normal (%)	39.8%	47.4%	35.1%	56.6%	50.0%	50.4%	43.3%	54.4%	57.1%	61.8%
Inhibited (%)	23.9%	25.9%	29.7%	22.6%	30.3%	27.9%	28.4%	25.0%	24.9%	21.5%

*Note*. *w GA* = weeks of gestation. *PIRI* = parent–infant relationship index. The following categorization was used for verbal inhibition responses: <180 s = disinhibited; 180–227 s = normal; >227 s = inhibited.

**Table 6 children-06-00081-t006:** Multinomial logistic regression results for gestational age group predicting verbal response compared to the full-term group.

	Variable	*B*	SE *B*	Wald	df	OR	95% CI
Disinhibited 180 s	Intercept	−1.09	0.20	29.39	1		
	Sex (male)	0.40 **	0.14	7.78	1	1.49	1.13–1.98
	Lower SES	−0.12 **	0.05	6.93	1	0.89	0.81–0.97
	Very preterm	0.93 ***	0.19	23.31	1	2.54	1.74–3.70
	Moderately preterm	0.71 *	0.28	6.37	1	2.03	1.17–3.52
	Late preterm	0.38	0.22	3.08	1	1.46	0.96–2.22
	Early term	0.48 *	0.21	5.04	1	1.61	1.06–2.44
Inhibited >227 s	Intercept	−1.57 ***	0.21	58.24	1		
	Sex (male)	0.01	0.14	0.00	1	1.01	0.77–1.32
	Lower SES	0.16 ***	0.05	12.87	1	1.18	1.08–1.29
	Very preterm	0.41 *	0.20	4.32	1	1.50	1.02–2.20
	Moderately preterm	0.34	0.28	1.47	1	1.40	0.81–2.42
	Late preterm	0.44 *	0.19	5.27	1	1.56	1.07–2.28
	Early term	0.34	0.20	2.82	1	1.40	0.95–2.07
	*LR χ^2^*	67.02 ***					

*Note:* * *p* < 0.05, ** *p* < 0.01, *** *p* < 0.001. The reference category is “normal” (180–227 s) verbal response. *n* = 1314.

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
