# Peer review of "Effects of Gestational Age and Early Parenting on Children’s Social Inhibition at 6 Years"

_children, 2019, doi:10.3390/children6070081_

Round 1
Reviewer 1 Report
[Children] Manuscript ID: children-507377
Thank you for the opportunity to review this interesting manuscript on peri- and postnatal effects on children’s social inhibition. The study assessed the impact of gestational age and parent-infant relationship on children’s social inhibition at age 6. Results indicated that children with lower gestational age were both more socially inhibited and disinhibited. Good parent-child relationships seemed to be a protective factor for verbal disinhibition.
The study includes data from an impressive longitudinal sample. I have a few concerns about the analyses and the overarching research questions of this sample.
Abstract
No comments
Introduction
Lines 54ff: Mothers of preterm children rate their children more highly on temperamental negativity than mothers of full term infants – Could this effect be related to physiological problems? That is, I was overall wondering if results on self-reports might stem from a halo effect. If mother (or fathers) perceive that their child has more problems (you kind of mention this later that these families are faced with an overall more difficult start of their child in life), they could easily also see more problems on the behavioral side.
Methods
Lines 104ff: I am curious about the observational system: Where do the categories come from? It is pointed out that after 180s, the stranger asked the child if it wanted to play. Yet, the categories are only based on reaction time. Could it be that it rather is „social cue“ vs. „no social cue“ instead of time?
Additionally, if reaction time is used, it would be easy to use it as a continuous dependent variable (see below).
Table S1: The interviews and observations were coded into yes/no categories. It is mentioned that the sum score did not show a normal distributation. Was it considered to use a dimensional appraoch in the coding system?
Results
Table 1: In my opinion, the analasis for SES should be a χ2 test and no ANOVA as you compare categories.
Following my comment from the Methods section, I believe it would be better to use reaction time as a dependent variable in an ANOVA instead of limiting the informative value of this parameter.
Table 6: I might have missed something but I do not really understand the additional value of this analysis.
Discussion
Lines 220 ff & 294ff: You claim that an interaction effect was tested (early parenting protects children against the problems of early gestational age) – This should be tested differently by a moderator analysis, e.g. by including an interaction term in the regression analysis. Therefore, this analysis should be added (or pointed out more clearly in both the statistical analysis and results section).
Lines 238 & 250ff: A neurodevelopmental perspective is brought into the discussion. While this provides an interesting point of view, it is not yet fully clear how well it relates to the current study. In which way did the current study cover this topic?
Lines 288f: It is mentioned that early regulatory problems in preterm infants are more predictive of disorganized attachment (and possibly social (dis)inhibition?) than sensitive parenting. What does this mean for your results? How important is parenting?
Lines 306f: One limitation includes the recruitment between 1985 and 1986. This is one of my main concerns: Is this study a secondary analysis of earlier data? Did the study design not allow to publish these findings earlier? Please provide more information (e.g., in the Methods) to allow judgement how this study should be classified.
Author Response
Introduction
1. Lines 54ff: Mothers of preterm children rate their children more highly on temperamental negativity than mothers of full term infants – Could this effect be related to physiological problems? That is, I was overall wondering if results on selfreports might stem from a halo effect. If mother (or fathers) perceive that their child has more problems (you kind of mention this later that these families are faced with an overall more difficult start of their child in life), they could easily also see more problems on the behavioral side.
Thank you for this comment. Although we agree with the reviewer that it is possible that children’s preterm birth status and subsequent physiological problems may contribute to parents’ overestimation of their children’s behavioral problems, we would like to point out that the outcome of interest in our study (i.e., social inhibition) was measured with a standardized experimental procedure rated by trained coders, rather than with parent reports.
We consider this a significant strength of the study, which adds to the current literature by showing that temperamental (i.e., social inhibition) differences are evident in children with lower gestational age. Thus, our findings extent previous reports of preterm children’s behavioral differences, which may have been biased by parents’ potentially subjective perceptions.
Methods
2. Lines 104ff: I am curious about the observational system: Where do the categories come from? It is pointed out that after 180s, the stranger asked the child if it wanted to play. Yet, the categories are only based on reaction time. Could it be that it rather is “social cue“ vs. no “social cue“ instead of time?
Thank you for mentioning this. The categories were based on the distribution of the healthy control children’s responses to the standardized task. As shown in Figures 1 and 2, the vast majority (60%) of these children responded verbally within 180-227 seconds and nonverbally within 180-188 seconds. This response pattern corresponded with the experimental stimulus (i.e., the stranger inviting the child to play at 180 seconds), which in our view strengthens the conceptual validity of the chosen cutoffs, as it shows that normal nonverbal response time was within seconds of the stranger’s invitation to play, and normal verbal response time was less than a minute after the stranger’s invitation to play. Thus, the answer to your questions is yes, children were coded as not socially inhibited or disinhibited if they showed the expected developmentally-appropriate response to the social cue. Since the timing of the social cue is part of the standardized experimental protocol we chose individual response times as the raw dependent variable.
3. Additionally, if reaction time is used, it would be easy to use it as a continuous dependent variable (see below).
We agree, and this is how we started our preliminary data analyses. However, because the distribution of response times was not normal, and because extremes on either end of the response-time continuum appeared to represent abnormal behavior (based on the healthy controls’ distributions), we determined that a categorical variable with cutoffs that corresponded to the experimental stimulus (i.e., the social cue) would more appropriately and meaningfully represent variation in the social inhibition construct.
4. Table S1: The interviews and observations were coded into yes/no categories. It is mentioned that the sum score did not show a normal distribution. Was it considered to use a dimensional approach in the coding system? Because most parents (52%) reported no (0) concerns in the parent-infant relationship, and to be consistent with previous work from the same cohort (e.g., Breeman, Jaekel, Baumann, Bartmann, & Wolke, 2017), we opted to maintain the binary coding approach. In our field, sum scores of different risks are considered highly reliable and valid markers, because they integrate information from different dimensions and sources of information, instead of considering individual dimensions separately (which would decrease statistical power and increase multicollinearity).
Results
5. Table 1: In my opinion, the analysis for SES should be a χ2 test and no ANOVA as you compare categories. Based on the distribution of the SES variable and its range from 1 to 6 (which we’ve now clarified in the manuscript in lines 93-95), we are using it as an interval scaled (continuous) variable in our main models. We believe this is additionally justified when considering the range of the standard deviation of this variable in each GA group and overall, as well as the otherwise considerably large loss of statistical power when dummy-coding 6 categories.
6. Following my comment from the Methods section, I believe it would be better to use reaction time as a dependent variable in an ANOVA instead of limiting the informative value of this parameter.
We would like to point out that because extremes on either

Reviewer 2 Report
I appreciate the opportunity to review this manuscript. This study evaluated the prospective relations preterm birth and later social inhibition and disinhibition in a large sample of children. Moreover, the study examined whether early parenting moderated the effects of preterm birth on social functioning. Results showed that lower gestational age was related to more social inhibition and disinhibition. Moreover, better early parent-infant relationships were associated with less verbal disinhibition. The study has several strengths including being a longitudinal study in a large and representative sample. Moreover, the study uses independent and objective measures for each construct. The study examines an important topic and the results of this study have the potential to make a valuable contribution in informing the development of temperament. However, I have several questions and suggestions that I believe would improve the manuscript.
1. The authors do not report reliability of the measures. Please report the reliability of the PIRI and the behavioral coding.
2. Please report the correlation between verbal and non-verbal responses. If they are correlated, I wonder if they should be combined into a single measure. Unless the authors have specific predictions for verbal vs. non-verbal indicators of social inhibition.
3. It is unclear if the PIRI was evaluated more than once per participant. Please clarify.
4. Several of the cited studies regarding children’s temperament also coded emotion expressions during temperament assessment. Have the authors considered this? Even if this would not be possible in such a large cohort, it should be discussed.
5. Similarly, recent evidence suggests that social inhibition may act as a developmental pathway between prenatal complications, including preterm birth, and later anxiety (Suarez, Morales, Metcalf, & Pérez-Edgar, 2019). If the authors have information regarding the psychopathology (e.g., anxiety) these children, it would greatly strengthen the manuscript.
6. I may be missing something, but I do not see where the interaction with parenting was statistically tested. However, in the discussion the authors mention a non-significant trend. Please report its associated statistics.
References:
Suarez, G. L., Morales, S., Metcalf, K., & Pérez-Edgar, K. E. (2019). Perinatal complications are associated with social anxiety: Indirect effects through temperament. Infant and Child Development, e2130.
Author Response
Responses to Reviewer 2 Comments
1. The authors do not report reliability of the measures. Please report the reliability of the PIRI and the behavioral coding.
Thank you for raising this point. We agree with the reviewer that more detailed information regarding the reliability of measures would enhance the report. For the social inhibition behavioral task developed by Jens Asendorpf (1993), inter-rater reliability was .93-.94, and re-test results showed high stability of .75 between ages 4 and 6 years. We have added this information to the manuscript in lines 107-109, which now read as follows:
The task was designed to specifically test children’s temperamental social inhibition and had high inter-rater (.93-.94) and re-test (.75 between ages 4 and 6 years) reliability in the original study [10].
For the PIRI, inter-rater reliability was not feasible to be assessed, since infants’ individually assigned nurses were trained for interviews and observations at each participating NICU.
2. Please report the correlation between verbal and non-verbal responses. If they are correlated, I wonder if they should be combined into a single measure. Unless the authors have specific predictions for verbal vs. non-verbal indicators of social inhibition.
Pearson correlation for the verbal and non-verbal response times was .91 and significant at the .01 level. However, because previous literature has shown that developmentally at-risk children may experience different patterns of verbal vs. nonverbal communication during dyadic social interaction with caregivers (Doussard-Roosevelt, Joe, Bazhenova, & Porges, 2003; Jaekel, Wolke, & Chernova, 2012) and peers (Agaliotis & Kalyva, 2008), we considered it important to separately explore their non-verbal and verbal responses to an adult stranger, in order to detect potential differences in the type of social approach. Nonetheless, our results showed that gestational age was related to differences in both verbal and non-verbal response patterns to the stranger.
References:
Agaliotis, I., & Kalyva, E. (2008). Nonverbal social interaction skills of children with learning disabilities. Research in Developmental Disabilities, 29(1), 1-10.
Doussard-Roosevelt, J. A., Joe, C. M., Bazhenova, O. V., & Porges, S. W. (2003). Mother–child interaction in autistic and nonautistic children: Characteristics of maternal approach behaviors and child social responses. Development and Psychopathology, 15(2), 277-295.
Jaekel, J., Wolke, D., & Chernova, J. (2012). Mother and child behaviour in very preterm and term dyads at 6 and 8 years. Developmental Medicine & Child Neurology, 54(8), 716-723.
3. It is unclear if the PIRI was evaluated more than once per participant. Please clarify.
We apologize for the confusion. Although the PIRI contained items that were assessed at different points in the infants’ development (i.e., neonatally, at 5 months), the total PIRI score was only obtained once per child, after all items had been combined. We have made this clearer in lines 102-103 which now read as follow:
After all items were obtained, the sum of responses was calculated and recoded into one binary variable (0=no concerns, 1=some degree of concern).
4. Several of the cited studies regarding children’s temperament also coded emotion expressions during temperament assessment. Have the authors considered this? Even if this would not be possible in such a large cohort, it should be discussed.
We appreciate this interesting point and agree with the reviewer that assessing emotion expressions during the social inhibition task would have provided important additional information. Unfortunately, although the task had been originally video-recorded, the tapes had been destroyed prior to analyses and there was no other documentation regarding affect or expression beyond latencies of social response. Nonetheless, we would like to point out that observers were trained to code only prosocial reactions (e.g., smiling, saying “hello”, etc.) when indicating latencies of children’s responses, as opposed to other non-verbal or verbal reactions. Moreover, the experimental task and coding were based on Asendorpf’s (1993) procedure, which was especially designed to assess temperament-based inhibition towards unfamiliar social stimuli. Accordingly, we adhere to descriptions of results as relating to social inhibition specifically, rather than temperament more broadly. Per the reviewer’s comment, however, we now mention the importance of considering additional indicators of temperamental traits in lines 350-352, which read as follows:
Future studies could consider coding additional indicators of temperamental approach tendencies (e.g., emotional expressions), in addition to social response time during experimental tasks of children’s interaction with adult strangers.
5. Similarly, recent evidence suggests that social inhibition may act as a developmental pathway between prenatal complications, including preterm birth, and later anxiety (Suarez, Morales, Metcalf, & Pérez-Edgar, 2019). If the authors have information regarding the psychopathology (e.g., anxiety) these children, it would greatly strengthen the manuscript.
Thank you for this comment and the excellent reference related to our work. We agree with the reviewer that exploring social inhibition as a developmental pathway to later social and mental health problems is a critical future direction that would further shed light on preterm individuals’ long-term outcomes. However, considering the lack of studies that explore social inhibition specifically in preterm children, as well as the novel findings from our current study regarding social disinhibition, we considered it important to establish both of these results before exploring more complex mediating mechanisms. Nonetheless, in acknowledgement of the reviewer’s important point, we now more explicitly mention the need to explore social inhibition as a potential mediator for later psychopathology, and we cite the provided reference in lines 360-362, which read as follow:
Additionally, future studies that explore social inhibition as a developmental pathway to preterm individuals’ later psychopathology [96] and social problems [97] constitute an important direction to shed light on the mechanisms explaining preterm individuals’ long-term outcomes.
6. I may be missing something, but I do not see where the interaction with parenting was statistically tested. However, in the discussion the authors mention a non-significant trend. Please report its associated statistics.
Thank you for this comment. Because the interaction was non-significant, we chose to report the results of the more parsimonious model without the interaction variable in the manuscript. However, per the reviewer’s request, we now include the results of the model with the interaction term in the supplementary material (Tables S4 and S5) and refer the reader to these results in the manuscript in lines 187-191, which now read as follows:
Because there was not a significant interaction effect of parent-infant relationship with gestational age on social inhibition, results reported above are only for the more parsimonious models without the non-significant interaction term. Results for the models with the interaction term can be found in Tables S4 and S5 in the Supplementary Material.
We describe the results provided in Table 5 as the non-significant trend.

Round 2
Reviewer 1 Report
[Children] Manuscript ID: children-507377
Thank you for this revised manuscript.
Overall, the manuscript has improved. Howeer, several of the answers should be introduced into the manuscript as well.
Previous comment 1: It is true that social inhibition was one of the outcome variables and objectively coded. However, parent-infant relationship (PIRI) was an additional outcome which was subjectively rated. Please include this at least into the limitations as the previously mentiones halo-effect cannot be excluded.
Previous comment 2 (categories social cue vs. No social cue) & 3 (reaction time as continuous variable): Thank you for your explanation. I would appreciate if you could introduce this information into the manuscript as I would expect other readers to be curious about these points as well.
Previous comment 11: While it may be that these results have not previously been published, my concerns stemmed from the fact that the hypotheses were not defined before assessment. As such, please provide information that the hypotheses were established post-hoc after assessment.
Reviewer 2 Report
I appreciate the authors being responsive to the reviewer’s comments and feedback. I believe the manuscript is improved and I do not have any further comments.
Author Response
We would like to thank the reviewer for this feedback.